# Beyond Individualized Recourse: Interpretable and Interactive Summaries of Actionable Recourses

**Kaivalya Rawal**
Harvard University
kaivalyarawal45@gmail.com

**Himabindu Lakkaraju**
Harvard University
hlakkaraju@seas.harvard.edu

## Abstract

As predictive models are increasingly being deployed in high-stakes decision-making, there has been a lot of interest in developing algorithms which can provide recourses to affected individuals. While developing such tools is important, it is even more critical to analyse and interpret a predictive model, and vet it thoroughly to ensure that the recourses it offers are meaningful and non-discriminatory *before it is deployed* in the real world. To this end, we propose a novel model agnostic framework called Actionable Recourse Summaries (AReS) to construct global counterfactual explanations which provide an interpretable and accurate summary of recourses for the entire population. We formulate a novel objective which simultaneously optimizes for correctness of the recourses and interpretability of the explanations, while minimizing overall recourse costs across the entire population. More specifically, our objective enables us to learn, with optimality guarantees on recourse correctness, a small number of compact rule sets each of which capture recourses for well defined subpopulations within the data. We also demonstrate theoretically that several of the prior approaches proposed to generate recourses for individuals are special cases of our framework. Experimental evaluation with real world datasets and user studies demonstrate that our framework can provide decision makers with a comprehensive overview of recourses corresponding to any black box model, and consequently help detect undesirable model biases and discrimination.

## 1 Introduction

Over the past decade, machine learning (ML) models are being increasingly deployed to make a variety of consequential decisions ranging from hiring decisions to loan approvals. Consequently, there is growing emphasis on designing tools and techniques which can explain the decisions of these models to the affected individuals and provide a means for *recourse* [38]. For example, when an individual is denied loan by a credit scoring model, he/she should be informed about the reasons for this decision and what can be done to reverse it. Several approaches in recent literature tackled the problem of providing recourses to affected individuals by generating *local* (instance level) counterfactual explanations [39, 36, 12, 27, 20]. For instance, Wachter et al. [39] proposed a model-agnostic, gradient based approach to find a closest modification (counterfactual) to any data point which can result in the desired prediction.

While prior research has focused on providing counterfactual explanations (recourses) for individual instances, it has left a critical problem unadressed. It is often important to analyse and interpret a model, and vet it thoroughly to ensure that the recourses it offers are meaningful and non-discriminatory *before it is deployed* in the real world. To achieve this, appropriate stake holders and decision makers should be provided with a high level, global understanding of model behaviour. However, existing techniques cannot be leveraged here as they are only capable of auditing individual instances. Consequently, while existing approaches can be used to provide recourses to affected

individuals after a model is deployed, they cannot assist decision makers in deciding if a model is good enough to be deployed in the first place.

**Contributions**: In this work, we propose a novel model agnostic framework called Actionable Recourse Summaries (AReS) to learn global counterfactual explanations which can provide an interpretable and accurate summary of recourses for the entire population with emphasis on specific subgroups of interest. These subgroups can be characterized either by specific features of interest input by an end user (e.g., race, gender) or can be automatically discovered by our framework. The goal of our framework is to enable decision makers to answer big picture questions about recourse–e.g.,*how does the recourse differ across various racial subgroups?*. To the best of our knowledge, this is the first work to address the problem of providing accurate and interpretable summaries of recourses which in turn enable decision makers to answer the aforementioned big picture questions.

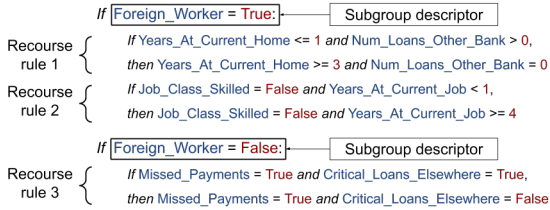

Figure 1: Recourse summary generated by our framework AReS. The outer-if rules describe the subgroups; the inner if-then rules are the recourse rules–recourse for an instance that satisfies the "if" clause is given by the "then" clause.

To construct the aforementioned explanations, we formulate a novel objective function which simultaneously optimizes for correctness of the recourses and interpretability of the resulting explanations, while minimizing the overall recourse costs across the entire population. More specifically, our objective enables us to learn, with optimality guarantees on recourse correctness, a small number of compact rule sets each of which capture recourses for well defined subpopulations within the data. We also demonstrate theoretically that several of the prior approaches proposed to generate recourses for individuals are special cases of our framework. Furthermore, unlike prior research, we do not make the unrealistic assumption that we have access to real valued recourse costs. Instead, we develop a framework which leverages Bradley-Terry model to learn these costs from pairwise comparisons of features provided by end users.

We evaluated the effectiveness of our framework on three real world datasets: credit scoring, judicial bail decisions, and recidivism prediction. Experimental results indicate that our framework outputs highly interpretable and accurate global counterfactual explanations which serve as concise overviews of recourses. Furthermore, while the primary goal of our framework is to provide high level overviews of recourses, experimental results demonstrate that our framework performs on par with state-of-the-art baselines when it comes to providing (instance level) recourses to affected individuals. Lastly, results from a user study we carried out suggest that human subjects are able to detect biases and discrimination in recourses very effectively using the explanations output by our framework.

**Related Work**: A variety of post hoc techniques have been proposed to explain complex models [8, 30, 15]. For instance, LIME [29] and SHAP [23], are *model-agnostic*, *local explanation* approaches which learn a linear model locally around each prediction. Other *local explanation* methods capture feature importances by computing the gradient with respect to the input [32, 34, 31, 33]. An alternate approach is to provide a global explanation summarizing the black box as a whole [17, 3], typically using an interpretable model. However, none of the aforementioned techniques were designed to learn counterfactual explanations or provide recourse.

Several approaches in recent literature addressed the problem of providing recourses to affected individuals by learning local counterfactual explanations for binary classification models [39, 27, 12, 36]. The main idea behind all these approaches is to determine what is the most desirable change that can be made to an individual's feature vector to reverse the obtained predictionDandl et al. [7]. Counterfactual explanations have also been leveraged to help in unsupervised exploratory data analysis of datasets in low dimensional latent spaces [26]. Wachter et al. [39], the initial proponents of counterfactual explanations for recourse, used gradient based optimization to search for the closest counterfactual instance to a given data point. Other approaches utilized standard SAT solvers [12], explanations output by methods such as SHAP [28], the perturbations in latent space found by autoencoders [25, 11], or the inherent structures of tree-based models [35, 22] to generate recourses.

More recent work focused on ensuring that the recourses being found were actionable. While Ustun et al. [36] proposed an efficient integer programming approach to obtain *actionable* recourses for linear classifiers, few other approaches focused on actionable recourses for tree based ensembles [35, 22]. Furthermore, Looveren and Klaise [20] and Poyiadzi et al. [27] proposed methods for obtaining more realistic counterfactuals by either prioritizing counterfactuals similar to certain class prototypes or ensuring that the path between the counterfactual and the original instance is one of high kernel density. Another way to build feasibility into counterfactual explanations is to suggest multiple counterfactuals for each data point, as done by Mothilal et al. [24] using gradient descent and Dandl et al. [7] using genetic algorithms. Although there are diverse techniques for finding counterfactual explanations, using these as recourses in the real-world is non-trivial since this involves making causal assumptions that may be violated when minimising recourse costs [13, 2, 37].

It is important to note that none of the aforementioned approaches provide a high level, global understanding of recourses which can be used by decision makers to vet the underlying models. Ours is the first work to address this problem by providing accurate and interpretable summaries of actionable recourses.

## 2 Our Framework: Actionable Recourse Summaries

Here, we describe our framework, Actionable Recourse Summaries (AReS), which is designed to learn global counterfactual explanations which provide an interpretable and accurate summary of recourses for the entire population. Below, we discuss in detail: 1) the representation that we choose to construct our explanations, 2) how we quantify the notions of fidelity, interpretability, and costs associated with recourses, 3) our objective function and its characteristics, 4) theoretical results demonstrating that our framework subsumes several of the previously proposed recourse generation approaches and, 5) optimization procedure with theoretical guarantees on recourse correctness.

### 2.1 Our Representation: Two Level Recourse Sets

The most important criterion for choosing a representation is that it should be understandable to decision makers who are not experts in machine learning, approximate complex non-linear decision surfaces accurately, and allow us to easily incorporate user preferences. To this end, we propose a new representation called two level recourse sets which builds on the previously proposed two level decision sets [17]. Below, we define this representation formally.

A **recourse rule** $r$ is a tuple $r = (c, c')$ embedded within an if-then structure i.e., "if $c$, then $c'$". $c$ and $c'$ are conjunctions of predicates of the form "feature $\sim$ value", where $\sim \in \{=, \geq, \leq\}$ is an operator (e.g., age $\geq 50$). Furthermore, there are two constraints that need to be satisfied by $c$ and $c'$: 1) the corresponding features in $c$ and $c'$ should match exactly and, 2) there should be at least one change in the values and/or operators between $c$ and $c'$. A recourse rule intuitively captures the current state ($c$) of an individual or a group of individuals, and the changes required to the current state ($c'$) to obtain a desired prediction. Figure 1 shows examples of recourse rules. A **recourse set** $S$ is a set of unordered recourse rules i.e., $S = \{(c_1, c_1'), (c_2, c_2') \cdots (c_L, c_L')\}$.

A **two level recourse set** $R$ is a hierarchical model consisting of multiple recourse sets each of which is embedded within an outer if-then structure (Figure 1). Intuitively, the outer if-then rules can be thought of as *subgroup descriptors* which correspond to different subpopulations within the data, and the inner if-then rules are *recourses* for the corresponding subgroups. Formally, a two-level recourse set is a set of triples and has the following form: $R = \{(q_1, c_{11}, c_{11}'), (q_1, c_{12}, c_{12}') \cdots (q_2, c_{21}, c_{21}') \cdots \}$ where $q_i$ corresponds to the subgroup descriptor, and $(c_{ij}, c_{ij}')$ together represent the inner if-then recourse rules with $c_{ij}$ denoting the antecedent (i.e., the if condition) and $c_{ij}'$ denoting the consequent (i.e., the recourse). A two level recourse set can be used to provide a recourse to an instance $x$ as follows: if $x$ satisfies exactly one of the rules $i$ i.e., $x$ satisfies $q_i \wedge c_i$, then its recourse is $c_i'$. If $x$ satisfies none of the rules in $R$, then $R$ is unable to provide a recourse to $x$. If $x$ satisfies more than one rule in $R$, then its recourse is given by the rule that has the highest probability of providing a correct recourse. Note that this probability can be computed directly from the data. Other forms of tie-breaking functions can be easily incorporated into our framework.

Table 1: Metrics used in the Optimization Problem

| Recourse Correctness | $\textbf{incorrectrecourse}(R) = \sum\limits_{i=1}^{M} |\{x|x \in \mathcal{X}_{\text{aff}}, x \text{ satisfies } q_i \wedge c_i, B(substitute(x, c_i, c_i')) \neq 1\}|$ |
|---|---|
| Recourse Coverage | $\textbf{cover}(R) = |\{x \mid x \in \mathcal{X}_{\text{aff}}, x \text{ satisfies } q_i \wedge c_i \; \exists i \in \{1 \cdots M\}\}|$ |
| Recourse Costs | $\textbf{featurecost}(R) = \sum\limits_{i=1}^{M} cost(c_i); \quad \textbf{featurechange}(R) = \sum\limits_{i=1}^{M} magnitude(c_i, c_i')$ |
| Interpretability | $\textbf{size}(R) = $ number of triples $(q, c, c')$ in $R$; $\textbf{maxwidth}(R) = \max\limits_{e \in \bigcup\limits_{i=1}^{M} (q_i \cup c_i)} num\_of\_predicates(e)$ <br><br> $\textbf{numrsets}(R) = |rset(R)|$ where $rset(R) = \bigcup\limits_{i=1}^{M} q_i$ |

## 2.2 Quantifying Recourse Correctness, Coverage, Costs, and Interpretability

To correctly summarize recourses for various subpopulations of interest, it is important to construct an explanation that not only accounts for the correctness of recourses, but also provides recourses to as many affected individuals as possible, minimizes recourse costs, and is interpretable. Below we explore each of these desiderata in detail and discuss how to quantify them w.r.t a two level recourse set $R$ with $M$ triples, a black box model $B$, and a dataset $\mathcal{X} = \{x_1, x_2 \cdots x_N\}$ where $x_i$ captures the feature values of instance $i$. We treat the black box model $B$ as a function which takes an instance $x \in \mathcal{X}$ as input and returns a class label–positive (1) or negative (0). We use $\mathcal{X}_{\text{aff}} \subseteq \mathcal{X}$ to denote those instances for which $B(x) = 0$ i.e., $\mathcal{X}_{\text{aff}}$ denotes the set of affected individuals who have received unfavorable predictions from the black box model.

**Recourse Correctness**: The explanations that we construct should capture recourses accurately. More specifically, when we use the recourse rules outlined by our explanation to prescribe recourses to affected individuals, they should be able to obtain the desired predictions from the black box model upon acting on the recourse. To quantify this notion, we define **incorrectrecourse**$(R)$ which is defined as the number of instances in $\mathcal{X}_{\text{aff}}$ for which acting upon the recourse prescribed by $R$ does not result in the desired prediction (Table 1). Our goal would therefore be to construct explanations that minimize this metric.

**Recourse Coverage**: It is important to ensure that the explanation that we construct *covers* as many individuals as possible i.e., provides recourses to them. To quantify this notion, we define **cover**$(R)$ as the number of instances in $\mathcal{X}_{\text{aff}}$ which satisfy the condition $q \wedge c$ associated with some rule $(q, c, c')$ in $R$ and are thereby provided a recourse by the explanation $R$ (Table 1).

**Interpretability**: The explanations that we output should be easy to understand and reason about. While choosing an interpretable representation (e.g., two level recourse sets) contributes to this, it is not sufficient to ensure interpretability. For example, while a decision tree with a hundred levels is technically readable by a human user, it cannot be considered as interpretable. Therefore, it is important to not only have an intuitive representation but also to achieve smaller complexity.

We quantify the interpretability of explanation $R$ using the following metrics: **size($R$)** is the number of triples of the form $(q, c, c')$ in the two level recourse set $R$. **maxwidth($R$)** is the maximum number of predicates (e.g., age $\geq 50$) in conjunctions $q \wedge c$ computed over all triples $(q, c, c')$ of $R$. **numrsets($R$)** counts the number of unique subgroup descriptors (outer if-then clauses) in $R$. All these metrics are formally defined in Table 1.

**Recourse Costs**: When constructing explanations, we also need to account for minimizing the recourse costs across the entire population. We define two metrics to quantify the recourse costs. First, we assume each feature is associated with a cost that captures how difficult it is to change that feature. This encapsulates the notion that some features are more *actionable* than the others. We define the total feature cost of a two-level recourse set $R$, **featurecost**$(R)$, as the sum of the costs of each of the features present in $c$ and whose value changes from $c$ to $c'$, computed across all triples $(q, c, c')$ of $R$. Second, apart from feature costs, it is also important to account for reducing the magnitude of changes in feature values. For example, it is much easier to increase income by 5K than 50K. To capture this notion, we define **featurechange**$(R)$ as the sum of *magnitude* of changes in feature values from $c$ to $c'$, computed across all triples in $R$. In case of categorical features, going from one value to another corresponds to a change of magnitude 1. Continuous features can be converted into ordinal features by binning the feature values. In this case, going from one bin to the next immediate bin corresponds to a change of magnitude 1. Other notions of magnitude can also be

easily incorporated into our framework. Our goal would be construct explanations that minimize the aforementioned metrics. Table 1 captures the formal definitions of these metrics.

### 2.2.1 Learning Feature Costs from Pairwise Feature Comparisons

One of the biggest challenges of computing **featurecost**$(R)$ is that it is non-trivial to obtain costs that capture the difficulty associated with changing a feature. For instance, even experts would find it hard to put precise costs indicating how *unactionable* a feature is. On the other hand, it is relatively easy for experts to make pairwise comparisons between features and assess which features are easier to change [6]. For example, a expert would be able to easily identify that it is much harder to increase income compared to number of bank accounts for any customer. Furthermore, there might be some ambiguity when comparing certain pairs of features and experts might have different opinions about which features are more actionable (e.g., increasing income vs. buying a car). So, it is important to account for this uncertainty .

AReS requires as input a vector of costs representing the difficulty of changing each model feature. We propose to learn this probabilistically in order to account for the variation in the opinions of experts regarding actionability of features. It is important to note however that AReS, is flexible enough to support feature costs computed in any other manner, or even specified directly through user input. This customization makes our method more generic compared to other existing counterfactual methods.

**The Bradley-Terry Model**: We leverage pairwise feature comparison inputs to learn the costs associated with each feature. To this end, we employ the well known Bradley-Terry model [21, 4] which states: If $p_{ij}$ is the probability that feature $i$ is less actionable (harder to change) compared to feature $j$, then we can calculate this probability as $p_{ij} = \frac{e^{\beta_i}}{e^{\beta_i}+e^{\beta_j}}$ where $\beta_i$ and $\beta_j$ correspond to the costs of features $i$ and $j$ respectively. Note that $p_{i,j}$ can be computed directly from the pairwise comparisons obtained by surveying experts, as $p_{ij} = \frac{\text{number of } i>j \text{ comparisons}}{\text{total number of } i,j \text{ comparisons}}$. We can then retrieve the costs of the features by learning the MAP estimates of feature costs $\beta_i$ and $\beta_j$ [5, 10].

## 2.3 Learning Two Level Recourse Sets

We formulate an objective function that can jointly optimize for recourse correctness, coverage, costs, and interpretability of an explanation. We assume that we are given as inputs a dataset $\mathcal{X}$, a set of instances $\mathcal{X}_{\text{aff}} \subseteq \mathcal{X}$ that received unfavorable predictions (i.e., labeled as 0) from the black box model $B$, a candidate set of conjunctions of predicates (e.g., age $\geq$ 50 and gender = female) $\mathcal{SD}$ from which we can pick the subgroup descriptors, and another candidate set of conjunctions of predicates $\mathcal{RL}$ from which we can choose the recourse rules. In practice, a frequent itemset mining algorithm such as apriori [1] can be used to generate the candidate sets of conjunctions of predicates. If the user does not provide any input, both $\mathcal{SD}$ and $\mathcal{RL}$ are assigned to the same candidate set generated by apriori.

To facilitate theoretical analysis, the metrics defined in Table 1 are expressed in the objective function either as non-negative reward functions or constraints. To construct non-negative reward functions, penalty terms (metrics in Table 1) are subtracted from their corresponding upper bound values ($U_1$, $U_3$, $U_4$) which are computed with respect to the sets $\mathcal{SD}$ and $\mathcal{RL}$.

$$f_1(R) = U_1 - \textbf{incorrectrecourse}(R), \text{ where } U_1 = |\mathcal{X}_{\text{aff}}| * \epsilon_1; f_2(R) = \textbf{cover}(R);$$
$$f_3(R) = U_3 - \textbf{featurecost}(R), \text{ where } U_3 = C_{max} * \epsilon_1 * \epsilon_2;$$
$$f_4(R) = U_4 - \textbf{featurechange}(R), \text{ where } U_4 = M_{max} * \epsilon_1 * \epsilon_2$$

where $C_{max}$ and $M_{max}$ denote the maximum possible feature cost and the maximum possible magnitude of change over all features respectively. These values are computed from the data directly. $\epsilon_1$ and $\epsilon_2$ are as described below. The resulting optimization problem is:

$$\underset{\mathcal{R}\subseteq\mathcal{SD}\times\mathcal{RL}}{\arg\max} \sum_{i=1}^{4} \lambda_i f_i(R) \tag{1}$$

$$\text{s.t. } \textbf{size}(R) \leq \epsilon_1, \textbf{maxwidth}(R) \leq \epsilon_2, \textbf{numrsets}(R) \leq \epsilon_3$$

$\lambda_1 \cdots \lambda_4$ are non-negative weights which manage the relative influence of the terms in the objective. These can be specified by an end user or can be set using cross validation (details in experiments section). The values of $\epsilon_1, \epsilon_2, \epsilon_3$ are application dependent and need to be set by an end user.

**Theorem 2.1.** *The objective function in Eqn. 1 is non-normal, non-negative, non-monotone, sub-modular and the constraints of the optimization problem are matroids.*

*Proof (Sketch).* Non-negative functions and submodular functions are both closed under addition and multiplication with non-negative scalars. Each term $f_i(R)$ is non-negative by construction. $f_2(R)$ is submodular, and the remaining terms are modular (and therefore submodular). Since $\lambda_i \geq 0$, the objective is submodular. To show that the objective is non-monotone, it suffices to show that $f_i$ is non-monotone for some $i$. Let $A$ and $B$ be two explanations s.t. $A \subseteq B$ i.e., $B$ has at least as many recourse rules as $A$. By definition, $incorrectrecourse(A) \leq incorrectrecourse(B)$ which implies that $f_1(A) \geq f_1(B)$. Therefore, $f_1$ is non-monotone. See Appendix for a detailed proof. $\qquad\square$

**Theorem 2.2.** *If all features take on values from a finite set, then the optimization problem in Eqn. 1 can be reduced to the objectives employed by prior approaches which provide instance level counterfactuals for individual recourse.*

*Proof (Sketch).* Individual recourse is represented in AReS by $\epsilon_1 = 1$ and having $q \wedge c$ consist of the entire feature-vector of a particular data-point $x \in \mathcal{X}_{\text{aff}}$. Additonally setting $\mathcal{SD} = \{x\}$ ensures the generated triples $(q, c, c')$ represent instance level counterfactuals. Finally, setting hyperparameter values $\lambda_2 = \lambda_3 = 0$ and $\epsilon_2 = \epsilon_3 = \infty$ leaves us with an objective function of $\lambda_1 f_1(R) + \lambda_4 f_4(R)$, which performs the same recourse search for individual recourse as the objectives outlined in prior work [39, 12, 36]. See Appendix for a detailed proof. $\qquad\square$

**Optimization Procedure** While exactly solving the optimization problem in Eqn. 1 is NP-Hard [14], the specific properties of the problem: non-monotonicity, submodularity, non-normality, non-negativity and the accompanying matroid constraints allow for applying algorithms with provable optimality guarantees. We employ an optimization procedure based on approximate local search [19] which provides the best known theoretical guarantees for this class of problems (Pseudocode for this procedure is provided in Appendix). More specifically, the procedure we employ provides an optimality guarantee of $\frac{1}{k+2+1/k+\delta}$ where $k$ is the number of constraints and $\delta > 0$.

**Theorem 2.3.** *If the underlying model provides recourse to all individuals, then upper bound on the proportion of individuals in $\mathcal{X}_{\text{aff}}$ for whom AReS outputs an incorrect recourse is $(1 - \rho)$, where $\rho \leq 1$ is the approximation ratio of the algorithm used to optimize Eqn 1.*

*Proof (Sketch).* Let $\Sigma_{i=1}^4 \lambda_i f_i'(R) = \Omega'$ represent the objective for the recourse set where any arbitrary point $x$ gets correct recourse (i.e., **incorrectrecourse**$' = 0$), obtained using $\epsilon_1 = 1$ and $\lambda_2 = \lambda_3 = \lambda_4 = 0$. Analogously $\Omega \geq \Omega'$ represents the recourse set with maximal objective function value. By def. $\Sigma_{i=1}^4 \lambda_i f_i(R) = \Omega^{A\bar{R}eS} \geq \rho\Omega \geq \rho\Omega'$. Solving, we find **incorrectrecourse**$^{AReS} \leq$ **incorrectrecourse**$' + \frac{\Omega'(1-\rho)}{\lambda_1} = [0 + \epsilon_1 + \frac{\Sigma_{i=2}^4 \lambda_i f_i'(R)}{\lambda_1 |\mathcal{X}_{\text{aff}}|}](1 - \rho) = (1 - \rho)$. See Appendix for a detailed proof. $\qquad\square$

**Generating Recourse Summaries for Subgroups of Interest** A distinguishing characteristic of our framework is being able to generate recourse summaries for subgroups that are of interest to end users. For instance, if a loan manager wants to understand how recourses differ across individuals who are foreign workers and those who are not, the manager can provide this feature as input. Our framework then provides an accurate summary of recourses while ensuring that the subgroup descriptors only contain predicates comprising of these features of interest (Figure 1).

Our two level recourse set representation naturally allows us to incorporate user input when generating recourse summaries. When a user inputs a set of features that are of interest to him, we simply restrict the candidate set of predicates $\mathcal{SD}$ (See Objective Function) from which subgroup descriptors are chosen to comprise only of those predicates with features that are of interest to the user. This will ensure that the subgroups in the resulting explanations are characterized by the features of user interest.

## 3 Experiments

Here, we discuss the detailed experimental evaluation of our framework AReS. First, we assess the quality of recourses output by our framework and how they compare with state-of-the-art baselines. Next, we analyze the trade-offs between recourse accuracy and interpretability in the context of our framework. Lastly, we describe a user study carried out with 21 human subjects to assess if human users are able to detect biases or discrimination in recourses using our explanations.

| Algorithms | | Datasets | | | | | |
|---|---|---|---|---|---|---|---|
| | | COMPAS | | Credit | | Bail | |
| | | Recourse Acc | Mean FCost | Recourse Acc | Mean FCost | Recourse Acc | Mean FCost |
| DNN | AR-LIME | **99.40%** | 3.42 | 10.26 % | 5.08 | 92.23 % | 2.90 |
| | AR-KMeans | 57.76 % | 6.39 | 48.72 % | 2.50 | 87.98 % | 7.50 |
| | FACE | 73.71% | 4.48 | 50.32% | 2.48 | 91.37% | 8.43 |
| | **AReS** | 99.37% | **2.83** | **78.23%** | **2.23** | **96.81%** | **2.45** |
| RF | AR-LIME | 65.88 % | 6.34 | 26.33 % | 3.32 | 90.46 % | 2.87 |
| | AR-KMeans | 60.48 % | 5.31 | 16.00 % | 4.02 | 92.03 % | 7.02 |
| | FACE | 62.38% | 4.48 | 31.32% | 1.35 | 92.37% | 9.31 |
| | **AReS** | **72.43%** | **2.52** | **39.87%** | **1.09** | **97.11%** | **1.78** |
| LR | AR | **100%** | 5.41 | **100%** | 1.69 | **100%** | 8.07 |
| | FACE | 98.22% | 6.12 | 95.31% | 2.08 | 94.37% | 7.35 |
| | **AReS** | 99.53% | **4.02** | 99.61% | **1.28** | **100%** | **6.45** |

Table 2: Evaluating recourse accuracy (recourse acc.) and mean feature cost (mean fcost) of recourses output by AReS and other baselines on COMPAS (left), Credit (middle), and Bail (right) datasets; DNN: 3-layer Deep Neural Network, RF: Random Forest, LR: Logistic Regression.

**Datasets**: Our first dataset is the **COMPAS** dataset which was collected by ProPublica [18]. This dataset captures information about the criminal history, jail and prison time, and demographic attributes of 7214 defendants from Broward County. Each defendant in the data is labeled either as high or low risk for recidivism. Our second dataset is the **German Credit** dataset [9]. This dataset captures financial and demographic information (e.g., account information, credit history, employment, gender) as well as personal details of 1000 loan applicants. Each applicant is labeled either as a good or a bad customer. Our third dataset is the **bail decisions** dataset [16], comprising of information pertaining to 18876 defendants from an anonymous county. This dataset includes details about criminal history, demographic attributes, and personal information of the defendants; with each defendant labeled as either high or low risk for committing crimes when released on bail.

**Baselines**: While there is no prior work on generating global summaries of recourses, we compare the efficacy of our framework in generating individual recourses with the following state-of-the-art baselines: (i) Feasible and Actionable Counterfactual Explanations (FACE) [27] [27] (ii) Actionable Recourse in Linear Classification (AR) [36]. While FACE produces actionable recourses by ensuring that the path between the counterfactual and the original instance is feasible, AR (for linear models only) performs an exhaustive search for counterfactuals. We adapt AR to non-linear classifiers by first constructing linear approximations of these classifiers using: (a) LIME [29] which approximates non-linear classifiers by constructing linear models locally around each data point (AR-LIME) and, (b) k-means to segment the dataset instances into $k$ groups ($k$ selected by cross-validation) and building one logistic regression model per group to approximate the non-linear classifier.

**Experimental Setup**: We generate recourses for the following classes of models – deep neural networks (DNN) with 3, 5, and 10 layers, gradient boosted trees (GBT), random forests (RF), decision trees (DT), SVM, and logistic regression (LR). We consider these models as black boxes through out the experimentation. Due to space constraints, we present results only with 3-layer DNN, RF, and LR here while remaining results are included in the Appendix. We split our datasets randomly into train (50%) and test sets (50%). We use the train set to learn the black box models, and the test set to construct and evaluate recourses. Furthermore, our objective function involves minimizing feature costs which we learn from pairwise feature comparisons (Section 2.2.1). In our experiments, we simulate these pairwise feature comparison inputs by randomly sampling a probability for every pair of features $a$ and $b$, which then dictates how unactionable is feature $a$ compared to feature $b$. We employ a simple tuning procedure to set the $\lambda$ parameters (details in Appendix) and the constraint values are assigned as $\epsilon_1 = 20$, $\epsilon_2 = 7$, and $\epsilon_3 = 10$. Support threshold for Apriori rule mining algorithm is set to 1%.

**Evaluating the Effectiveness of Our Recourses**: To evaluate the accuracy and cost-effectiveness of the recourses output by our framework and other baselines, we outline the following metrics: 1) *Recourse Accuracy*: percentage of instances in $\mathcal{X}_{\text{aff}}$ for which acting upon the prescribed recourse (e.g., changing the feature values as prescribed by the recourse) obtains the desired prediction. 2) *Mean FCost*: Average feature cost computed across those individuals in $\mathcal{X}_{\text{aff}}$ for whom prescribed recourses resulted in desired outcomes. Feature cost per individual is the sum of the costs of those features that need to be changed (as per the prescribed recourse) to obtain the desired prediction. In case of our framework, if an instance satisfies more than one recourse rules, we consider only the rule that has the highest probability of providing a correct recourse (computed over all instances that

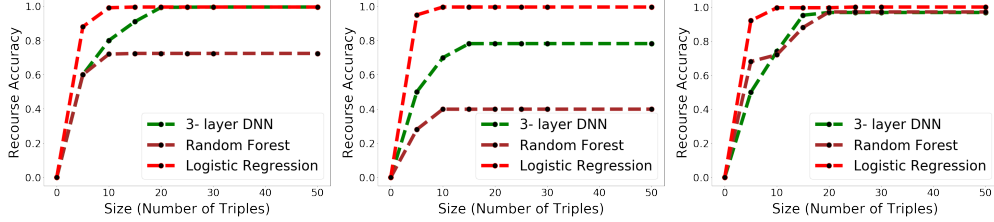

Figure 2: Analyzing the trade-Offs between interpretability and correctness of recourse: Size of the Explanation vs. Recourse Accuracy for COMPAS (left), Credit (middle), and Bail (right) datasets

satisfy the rule). In practice, however, we found that there are very few instances (between 0.5% to 1.5% across all datasets) that satisfy more than one recourse rule. We compute the aforementioned metrics on the recourses obtained by our framework and other baselines. Results with three different black box models are shown in Table 2.

It can be seen that there is no single baseline that performs consistently well across all datasets and all black boxes. For instance, AR-LIME performs very well with DNN on COMPAS (recourse accuracy of 99.4%) but has the least recourse accuracy on the credit dataset for the same black box (10.3%). Similarly, FACE results in low costs (mean fcost = 1.35) with RF on credit data but outputs very high cost recourses (mean fcost = 9.31) on bail data with the same black box. While AR performs consistently well in terms of recourse accuracy (100%) for LR black box mainly due to the fact that it uses coefficients from linear models directly to obtain recourses, it cannot be applied to any other non-linear model directly. On the other hand, AReS shows that is is possible to obtain recourses with low cost and high (individual) accuracy despite being model agnostic and operating on the subgroup level. Among our particular experiments, summarised in table 2, AReS always had the lowest FCost, and provided recourses that either had the highest Recourse Accuracy or were within 0.5% of the best performing individual recourse generation method.

**Analyzing the Trade-Offs Between Interpretability and Recourse Accuracy**: Since our framework optimizes both for recourse correctness and interpretability simultaneously, it is important to understand the trade-offs between these two aspects in the context of our framework. Note that this analysis is not applicable to any other prior work on recourse generation because previously proposed approaches solely focus on generating instance-level recourses and not global summaries, and thereby do not have to account for these tradeoffs. Here, we evaluate how the recourse accuracy metric (defined above) changes as we vary the *size* of the two level recourse set i.e., number of triples of the form $(q, c, c')$ in the two level recourse set (Section 2.2). Results for the same are shown in Figure 2. It can be seen that recourse accuracies converge to their maximum values at explanation sizes of about 10 to 15 rules across all the datasets. Since humans are capable of understanding and reasoning with rule sets of this magnitude [16], results in Figure 2 establish that we are not sacrificing recourse accuracy to achieve interpretability in case of the datasets or the black box models that we are using.

**Detecting Biases in Recourses - A User Study**: To evaluate if users are able to detect model biases or discrimination against specific subgroups using the recourse summaries output by our framework, we carried out an online user study with 21 participants. To this end, we first constructed as our black box model a two level recourse set that was biased against one racial subgroup i.e., it required individuals of one race to change twice the number of features to obtain a desired prediction (details in Appendix). We then used our framework AReS and AR-LIME (see Baselines) to construct the recourses corresponding to this black box. Note that while our method outputs global summaries of recourses, AR-LIME can only provide instance level recourses. However, since there is no prior work which provides global summaries of recourses like we do, we use AR-LIME and average its instance level recourses as discussed in Ustun et al. [36] and use it as a comparison point for this study. Participants were randomly assigned to see either the recourses output by our method (customized to show recourses for various racial subgroups) or AR-LIME. We also provided participants with a short tutorial on recourses as well as the corresponding methods they were assigned to. Participants were then asked two questions: 1) *Based on the recourses shown above, do you think the underlying black box is biased against a particular racial subgroup?* 2) *If so, please describe the nature of the bias in plain English.*. While the first question was a multiple choice question with three answer options: *yes, no, hard to determine*, the second question was a descriptive one.

We then evaluated the responses of all the 21 participants. Each descriptive answer was examined by two independent evaluators and tagged as right or wrong based on if the participant's answer accurately described the bias or not. We excluded from our analysis one response on which the evaluators did not agree. Our results show that 90% of the participants who were assigned to our method AReS were able to accurately detect that there is an underlying bias. Furthermore, 70% of these participants also described the nature of the bias accurately. On the other hand, out of the 10 participants assigned to AR-LIME, only two users (20%) were able to detect that there is an underlying bias and no user (0%) was able to describe the bias correctly. In fact, 80% of the participants assigned to AR-LIME said it was *hard to determine* if there is an underlying bias. These results clearly demonstrate the necessity and significance of methods which can provide accurate summaries of recourses as opposed to just individual recourses.

In addition to the above, we also experimented with introducing racial biases into a 3-layer neural network and a logistic regression model via trial and error. We then carried out similar user studies (as above) with 36 additional participants to evaluate how our explanations compared with aggregates of individual recourses. In case of the 3-layer neural network, AReS clearly outperformed AR-LIME – 88.9% vs. 44.4% on bias detection and 55.6% vs. 11.1% on bias description. In case of the logistic regression model, AReS and AR-LIME performed comparably – 88.9% in both cases on bias detection and 66.7% vs. 44.4% on bias description.

## 4    Conclusions

In this paper, we propose AReS, the first ever framework designed to learn global counterfactual explanations which can provide interpretable and accurate summaries of cost-effective recourses for the entire population with emphasis on specific subgroups of interest. Extensive experimentation with real world data from credit scoring and criminal justice domains as well as user studies suggest that our framework outputs interpretable and accurate summaries of recourses which can be readily used by decision makers and stakeholders to diagnose model biases and discrimination. This work paves way for several interesting future directions. First, the notions of recourse correctness, costs, and interpretability that we outline in this work can be further enriched. Our optimization framework can readily incorporate any newer notions as long as they satisfy the properties of non-negativity and submodularity. Second, it would be interesting to explore other real-world settings to which this work can be applied.

## 5    Broader Impact

Our framework, AReS, can be used by decision makers that wish to analyse machine learning systems for biases in recourse before deploying them in the real world. It can be applicable to a variety of domains where algorithms are making decisions and recourses are necessary–e.g., healthcare, education, insurance, credit-scoring, recruitment, and criminal justice. AReS enables the auditing of systems for fairness, and its customizable nature allows decision makers to specifically test and understand their models in a context dependent manner.

It is important to be cognizant of the fact that just like any other recourse generation algorithm, AReS may also be prone to errors. For instance, spurious recourses may be reported either due to particular configurations of hyperparameters (e.g., valuing coverage or interpretability way more than correctness) or due to the approximation algorithms we use for optimization. Such errors may translate into masking existing biases of a classifier, especially on subgroups that are particularly underrepresented in the data. It may also lead to AReS reporting nonexistent biases. It is thus important to be cognizant of the fact that AReS is finally an explainable algorithm (as opposed to being a fairness technique) that is meant to guide decision makers. It can be used to gauge the need for deeper analysis, and test for specific, known red-flags, rather than to provide concrete evidence of violations of fairness criteria.

Good use of AReS requires decision makers to be cognizant of these strengths and weaknesses. For a more complete understanding of a black box classifier before deployment, we recommend that AReS be run multiple times, with different hyperparameters and candidate sets $\mathcal{RL}$ and $\mathcal{SD}$. Furthermore, evaluating the interpretability-recourse accuracy tradeoffs (Figure 2) can help detect any undesirable scenarios which might result in spurious recourses. Possible violations of fairness criteria discovered

by AReS should be investigated further before any action is taken. Finally, we propose that when showing summaries output by AReS, the recourse accuracies of each of the recourse rules should also be included so that decision makers can make informed choices.

## Acknowledgements

We would like to thank Julius Adebayo, Winston Luo, Hayoun Oh, Sarah Tan, and Berk Ustun for insightful discussions. This work is supported in part by Google. The views expressed are those of the authors and do not reflect the official policy or position of the funding agencies.

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
