[Supplementary Material]

# A  Appendix

## A.1  Proofs for Theorems

**Theorem 2.1.** *The objective function in Equation 1 is non-normal, non-negative, non-monotone, submodular, and the constraints of the optimization problem are matroids.*

*Proof.* In order to prove that the objective function in Eqn. 1 is non-normal, non-negative, non-monotone, and submodular, we need to prove the following:

- any one of the terms in the objective is non-normal

- all the terms in the objective are non-negative

- any one of the terms in the objective is non-monotone

- all the terms in the objective are submodular

**Non-normality**    Let us consider the term $f_1(R)$. If $f_1$ is normal, then $f_1(\emptyset) = 0$.

It can be seen from the definition of $f_1$ that $f_1(\emptyset) = U_1$ because **incorrectrecourse**$(\emptyset) = 0$ by definition. This also implies that $f_1(\emptyset) \neq 0$. Therefore $f_1$ is non-normal and consequently the entire objective is non-normal.

**Non-negativity**    The functions $f_1, f_3, f_4$ are non-negative because first term in each of these is an upper bound on the second term. Therefore, each of these will always have a value $\geq 0$. In the case of $f_2$ which encapsulates the **cover** metric which is the number of instances which satisfy some recourse rule in the explanation. This metric can never be negative by definition. Since all the functions are non-negative, the objective itself is non-negative.

**Non-monotonicity**    Let us choose the term $f_1(R)$. Let us consider two explanations (two level recourse sets) $R_1$ and $R_2$ such that $R_1 \subseteq R_2$. If $f_1$ is monotonic then, $f_1(R_1) \leq f_1(R_2)$. Let us see if this condition holds:

Based on the definition of **incorrectrecourse** metric, it is easy to note that

$$incorrectrecourse(R_1) \leq incorrectrecourse(R_2)$$

This is because $B$ has at least as many rules as that of $A$. This implies the following:

$$-incorrectrecourse(R_1) \geq -incorrectrecourse(R_2)$$

$$U_1 - incorrectrecourse(R_1) \geq U_1 - incorrectrecourse(R_2)$$

$$f_1(R_1) \geq f_1(R_2)$$

This shows that $f_1$ is non-monotone and therefore the entire objective is non-monotone.

**Submodularity**    Let us go over each of the terms in the objective and show that each one of those is submodular.

Let us consider two explanations (two level recourse sets) $R_1$ and $R_2$ such that $R_1 \subseteq R_2$. A function $f$ is considered to be submodular if $f(R_1 \cup e) - f(R_1) \geq f(R_2 \cup e) - f(R_2)$ where $e = (q, c, c') \notin R_2$.

By definition of **incorrectrecourse**, each time a triple $(q, c, c')$ is added to some explanation $R$, the value of **incorrectrecourse** is simply incremented by the number of data points for which this triple assigns recourse incorrectly. This implies that this metric is modular which in turn means $f_1$ is also modular and thereby submodular i.e.,

$$f_1(R_1 \cup e) - f_1(R_1) = f_1(R_2 \cup e) - f_1(R_2)$$

$f_2$ is the cover metric which denotes the number of instances that satisfy some rule in the explanation. This is clearly a diminishing returns function i.e., more additional instances in the data are covered

when we add a new rule to a smaller two level recourse set compared to a larger one. Therefore, $f_2$ is submodular.

**featurecost** and **featurechange** are both additive in that each time a triple $(q, c, c')$ is added to some explanation $R$, the value of these metrics is incremented either by the sum of costs of corresponding features which need to be changed (**featurecost**) or the sum of changes in magnitudes of the features (**featurechange**). This implies that these metrics are modular which in turn means $f_3$ and $f_4$ are modular and thereby submodular.

**Constraints:** A constraint is a matroid if it has the following properties: 1) $\emptyset$ satisfies the constraint 2) if 2 two level recourse sets $R_1$ and $R_2$ satisfy the constraint and $|R_1| < |R_2|$, then adding an element $e = (q, c, c')$ s.t. $e \in R_2, e \notin R_1$ to $R_1$ should result in a set that also satisfies the constraint. It can be seen that these two conditions hold for all our constraints. For instance, if a two level recourse set $R_2$ has $\leq \epsilon_1$ rules (i.e., **size**$(R_2) \leq \epsilon_1$) and another two level recourse set $R_1$ has fewer rules than $R_2$, then the set resulting from adding any element of $R_2$ to the smaller set $R_1$ will still satisfy the constraint on **size**. Similarly, the constraints on **maxwidth** and **numrsets** satisfy the aforementioned properties too. $\qquad\square$

Before we prove Theorem 2.2, we will first discuss how several previously proposed methods which provide recourses for affected individuals (i.e., instance level recourses) can be unified into one basic algorithm.

### Unifying Prior Work

The algorithm below unifies multiple prior instance-level recourse finding techniques namely Wachter et al. [39], Ustun et al. [36], Karimi et al. [12]. All the aforementioned techniques employ a generalized optimization procedure that searches for a minimum cost recourse for every affected individual by constantly polling the classifier $B$ with different candidate recourses until a valid recourse is found [7]. The search for valid recourses is guided by the $find$ function, which generates candidates with progressively higher costs (with the definition of cost varying by technique). For example, Wachter et al. [39] use ADAM to optimize their cost function, $\lambda(B(x') - B(x))^2 + d(x', x)$ - where $d$ represents a distance metric (e.g., L1 norm), to repeatedly generate candidates for $x'$ increasingly farther away from $x$, until one of them finally flips the classifier prediction. Similarly, Karimi et al. [12] use boolean SAT solvers to exhaustively generate candidate modifications $x'$, while Ustun et al. [36] use integer programming to generate candidate modifications that are monotonically non-decreasing in cost, thus providing the theoretical guarantee of finding minimum cost recourse for linear models.

---

**Algorithm 1** Generalised Recourse Generation Procedure

---

1: **Input:** binary black-box classifier $B$, dataset $\mathcal{X}$, single data point $x$, iterator $find$ to repeatedly generate candidate modifications to $x$
2: **Result:** minimal feature-vector modifications $\Delta x$ needed for $B(x + \Delta x)$ to be different from $B(x)$, or $\emptyset$ if no recourse exists
3: $output = B(x)$
4: $\Delta x = \emptyset$
5: $x' = x + \Delta x$ $\qquad\qquad\qquad\triangleright$ $x'$ is the candidate counterfactual for $x$ with modification $\Delta x$
6: **while** $B(x') = output$ **do**
7: $\quad \Delta x = find(B, x, x', \mathcal{X})$ $\qquad\qquad\triangleright$ $find$ returns $\emptyset$ if it cannot find any candidate $\Delta x$
8: $\quad$ **if** $\Delta x = \emptyset$ **then**
9: $\quad\quad$ return $\emptyset$ $\qquad\qquad\qquad\qquad\qquad\qquad\qquad\triangleright$ No recourse found
10: $\quad$ **end if**
11: $\quad x' = x + \Delta x$
12: **end while**
13: **return** $\Delta x$

---

**Theorem 2.2** *If all features take on values from a finite set, then the optimization problem in Eqn.1 can be reduced to the objectives employed by prior approaches which provide instance level counterfactuals for individual recourse.*

*Proof.* To prove this theorem, we will first describe how instance level counterfactuals for indivual recourses can be generated using our framework AReS. Then, we show how this is equivalent to the objectives outlined in Wachter et al. [39], Ustun et al. [36], Karimi et al. [12].

*Generating instance level counterfactuals for individual recourses using AReS:* If a conjunction $q \wedge c$ consists of the entire feature-vector of a particular data-point $x \in \mathcal{X}_{\text{aff}}$, then the triple $(q, c, c')$ represents a single instance level counterfactual. This is how AReS can be used to output individual recourses.

*Subsuming other objective functions:* The objective optimized by Wachter et al. is $\lambda(B(x') - B(x))^2 + d(x', x)$. This can be equivalently expressed in our notation from Table 1 as $\lambda(\textbf{incorrectrecourse}(R)) + \textbf{featurechange}(R)$, where the first term captures how closely the prediction resulting from the prescribed recourse matches the desired prediction, and the second term represents the distance between the counterfactual and the original data point $x \in X_{\text{aff}}$. The aforementioned two expressions are equivalent because our setting consists only of binary classifiers with $0/1$ outputs. In this case, our definition of $\textbf{incorrectrecourse}(R) = \Sigma_{i=1}^{M}|\{x|x \in \mathcal{X}_{\text{aff}}, x \text{ satisfies } q_i \wedge c_i, B(\textit{substitute}(x, c_i, c_i')) \neq 1\}|$ is identical to $\textbf{incorrectrecourse}(R) = \Sigma_{i=1}^{M}(B(x) - B(x + \Delta c_i)))^2$. Similarly, $\textbf{featurechange}(R)$ from our notation is the same as $d(x', x)$. As described in algorithm 1, all recourse search techniques use the notion captured by $\textbf{incorrectrecourse}(R)$ and some form of distance metric or cost function, captured by $\textbf{featurechange}(R)$ or the customizable $\textbf{featurecost}(R)$ in AReS.

Let the (finite) set of all possible feature vectors be denoted by $\mathcal{X}_{\text{all}}$. Note that $\mathcal{X}_{\text{all}} \supseteq \mathcal{X}$, and setting $\mathcal{RL} = \mathcal{X}_{\text{all}}$ in AReS would allow the recourse search to be over the entire domain of the data. Setting $\textbf{size}(R) = 1$ and $\mathcal{SD} = \{x\}$ further mandates that the final recourse set consists of only one triple $(q, c, c')$, which contains the recourse desired for the feature-vector $x$. Further, since most instance level recourse generation techniques do not have additional interpretability constraints [39, 36, 27, 12, 20, 25] such as the $\textbf{maxwidth}(R)$ and $\textbf{numrsets}(R)$ terms in AReS, we set $\epsilon_2 = \epsilon_3 = \infty$. Finally, setting $\lambda_2 = \lambda_3 = 0$ leaves us with $\lambda_1 f_1(R) + \lambda_4 f_4(R)$ as our objective function, which represents the exact same optimization as that of Wachter et al. [39]. Similar configurations (e.g. setting $\lambda_4 = 0$ instead of $\lambda_3 = 0$, and defining cost of each feature in terms of percentile shift in feature values) will yield the objective functions used by other recourse generation techniques (e.g. Ustun et al.'s Actionable Recourse). $\square$

**Theorem 2.3** *If the underlying model provides recourse to all individuals, then upper bound on the proportion of individuals in $\mathcal{X}_{\text{aff}}$ for whom AReS outputs an incorrect recourse is $(1 - \rho)$, where $\rho \leq 1$ is the approximation ratio of the algorithm used to optimize Eqn 1.*

*Proof.* Let $\Sigma_{i=1}^{4} \lambda_i f_i(R^\Omega) = \Omega$ represent the maximum possible value of the objective function defined in Eqn. 1. Let $\Sigma_{i=1}^{4} \lambda_i f_i(R') = \Omega'$ represent the objective value for the two level recourse set which provides correct recourse to a single arbitrary data point $x$ (i.e., $\textbf{incorrectrecourse}' = 0$) which is obtained by setting $\epsilon_1 = 1$ and $\lambda_2 = \lambda_3 = \lambda_4 = 0$. Therefore, $\Omega \geq \Omega'$ and $\Sigma_{i=1}^{4} \lambda_i f_i(R) = \Omega^{AReS} \geq \rho\Omega$ due to the approximation ratio ($\rho \leq 1$) of the algorithm used to optimize Eqn. 1.

$$\sum_{i=1}^{4} \lambda_i f_i(R^\Omega) = \Omega \tag{2}$$

$$\sum_{i=1}^{4} \lambda_i f_i(R) \geq \rho\Omega \tag{3}$$

subtracting (3) from (2), we get

$$\sum_{i=1}^{4} \lambda_i(f_i(R^\Omega) - f_i(R)) \leq \Omega - \rho\Omega \tag{4}$$

| Algorithms | | Datasets | | | | | |
| --- | --- | --- | --- | --- | --- | --- | --- |
| | | **COMPAS** | | **Credit** | | **Bail** | |
| | | Recourse Accuracy | Mean FCost | Recourse Accuracy | Mean Fcost | Recourse Accuracy | Mean Fcost |
| DNN-5 | AR-LIME | **99.67%** | 2.93 | 0% | NA | 84.49% | 2.59 |
| | AR-KMeans | 65.89% | 6.07 | 47.06% | 1.68 | 92.25% | 7.31 |
| | FACE | 88.28% | 5.43 | 68.31% | 2.25 | 83.31% | 5.64 |
| | **AReS** | 98.72% | **1.92** | **83.02%** | **1.03** | **96.18%** | **1.88** |
| GBT | AR-LIME | 21.57% | 5.21 | 8.00% | 3.44 | 69.17% | 2.40 |
| | AR-KMeans | 60.08% | 5.34 | 22.33% | 3.40 | 93.03% | 7.14 |
| | FACE | 55.87% | 5.42 | 24.38% | 3.41 | 77.82% | 5.63 |
| | **AReS** | **76.17%** | **3.88** | **58.32%** | **1.67** | **97.84%** | **1.18** |
| SVM | AR | **100%** | 1.25 | **100%** | 7.84 | **100%** | 7.93 |
| | FACE | 95.63% | 1.43 | 93.10% | 5.77 | 88.12% | 7.02 |
| | **AReS** | 99.64 | **0.88** | **100%** | **2.45** | **100%** | **4.35** |

Table 3: Evaluating Recourse Accuracy and Mean FCost of recourses output by AReS and other baselines on COMPAS (left), Credit (middle), and Bail (right) datasets; DNN-5: 5 Layer Deep Neural Network, GBT: Gradient Boosted Trees, SVM: Support Vector Machine. Higher values of recourse accuracy are desired; lower values of mean fcost are desired.

Optimizing only for recourse correctness of a single arbitrary instance i.e., setting $\epsilon_1 = 1$ and $\lambda_2 = \lambda_3 = \lambda_4 = 0$, we have $\Omega \to \Omega'$. Therefore, Eqn. (4) can be written as:

$$\lambda_1(f_1(R') - f_1(R)) \leq \Omega'(1 - \rho)$$

$$(U_1 - \mathbf{incorrectrecourse}(R')) - (U_1 - \mathbf{incorrectrecourse}(R)) \leq \frac{\Omega'(1 - \rho)}{\lambda_1}$$

$$\mathbf{incorrectrecourse}(R) \leq 0 + \frac{\Omega' \times (1 - \rho)}{\lambda_1}$$

$$\mathbf{incorrectrecourse}(R) \leq 0 + \frac{\lambda_1(U_1 - 0) \times (1 - \rho)}{\lambda_1}$$

Using the definition of $U_1$ from Section 2.3, $\dfrac{\mathbf{incorrectrecourse}(R)}{|\mathcal{X}_{\mathrm{aff}}|} \leq \dfrac{\epsilon_1|\mathcal{X}_{\mathrm{aff}}| \times (1 - \rho)}{|\mathcal{X}_{\mathrm{aff}}|}$

$$\frac{\mathbf{incorrectrecourse}(R)}{|\mathcal{X}_{\mathrm{aff}}|} \leq (1 - \rho)$$

$\square$

This establishes that the upper bound on the proportion of individuals in $\mathcal{X}_{\mathrm{aff}}$ for whom AReS outputs an incorrect recourse is $(1 - \rho)$.

## A.2 Experimental Evaluation

### A.2.1 Parameter Tuning

We set the parameters $\lambda_1 \cdots \lambda_4$ as follows. First, we set aside 5% of the dataset as a validation set to tune these parameters. We first initialize the value of each $\lambda_i$ to 100. We then carry out a coordinate descent style approach where we decrement the values of each of these parameters while keeping others constant until one of the following conditions is violated: 1) less than 95% of the instances in the validation set are *covered* by the resulting explanation 2) more than 2% of the instances in the validation set are *covered* by multiple rules in the explanation 3) the prescribed recourses result in incorrect labels for more than 15% of the instances (for whom the black box assigned label 0) in the validation set.

Figure 3: Analyzing the trade-Offs between interpretability and correctness of recourse: Size of the Explanation vs. Recourse Accuracy for COMPAS (left), Credit (middle), and Bail (right) datasets

### A.2.2 User Study

We manually constructed a two level recourse set (as our black box model) for the bail application. We deliberately ensured that this black box was biased against individuals who are not Caucasian. More specifically, we induced the following bias: individuals who are not Caucasian are required to change twice the number of features to obtain a desired prediction compared to those who are Caucasian. This two level recourse set (black box) is shown in Figure 4.

We then used our approach and 95% of the bail dataset to learn a two level recourse set explanation (remaining 5% of the data is used for tuning $\lambda_1 \cdots \lambda_4$ parameters). We also set all feature costs to 1. We found that our approach was able to *exactly* recover the underlying model and thereby obtain a recourse accuracy of 100%. We used AR-LIME as a comparison point in our user study. Note that while our method outputs global summaries of recourses, AR-LIME can only provide instance level recourses. However, since there is no prior work which provides global summaries of recourses like we do, we use AR-LIME and average its instance level recourses as discussed in Ustun et al. [36]. More specifically, we first run AR-LIME to obtain individual recourses and then for each possible subgroup of interest, we will average the recourses over all individuals within that subgroup (as suggested in Ustun et al. [36]). We found that such an averaging was actually resulting in incorrect summaries which are misleading. This in turn reflected in the user responses of our user study.

Figure 4: Biased black box classifier that we constructed. Red colored feature-value pairs represent the changes that need to be made to obtain desired predictions. Note that individuals who are not Caucasian will need to change two of their feature values (e.g., *Property* and *Has Job*; *Drugs* and *Pays Rent*) to obtain the desired outcome. On the other hand, Caucasians only need to change one feature (e.g., *Has Job*; *Drugs*). Our framework AReS was able to recover the same exact model.

If Female =No and Foreign Worker =No:

      **If** Missed Payments =Yes **and** Critical Loans =Yes, **then** Missed Payments =Yes **and** Critical Loans =No

      **If** Unemployed =Yes **and** Critical Loans =Yes **and** Has Guarantor =No,

          **then** Unemployed =Yes **and** Critical Loans =No **and** Has Guarantor =Yes

**If** Female =No **and** Foreign Worker =Yes:

      **If** Skilled Job =No **and** Years at Job $\leq$ 1, **then** Skilled Job =Yes **and** Years at Job $\geq$ 4

      **If** Unemployed =Yes **and** Has Guarantor =No **and** Has CoAppplicant =No,

          **then** Unemployed =No **and** Has Guarantor =Yes **and** Has CoAppplicant =Yes

**If** Female =Yes:

      **If** Married =No **and** Owns House =No, **then** Married =Yes **and** Owns House =Yes

      **If** Unemployed =No **and** Has Guarantor =Yes **and** Has CoAppplicant =No,

          **then** Unemployed =No **and** Has Guarantor =Yes **and** Has CoAppplicant =Yes

Figure 5: Recourse summary generated by our framework AReS for a 3-layer DNN (black box) on credit scoring application. Red colored feature-value pairs represent the changes that need to be made to obtain desired predictions.