[Reviews · NeurIPS 2020]

Review 1

Summary and Contributions: The paper has identified an interesting gap in the recourse literature, specifically focusing on sub-population-based recourse. The authors propose a first method to solve this task.

Strengths: Overall, this is a well-rounded paper and provides a fresh perspective on a relevant problem: counterfactual explanations. Specifically, in my opinion, this paper's novelty is actionable recourse *summaries*, optimizing for group recourse rather than individual recourse, which they motivate well by providing an excellent introduction, a good mix of theory, experiments, and a brief user-study.

Weaknesses: My primary hesitation with this work is several over-reaching claims, and some unanswered questions. I personally wish to this see paper come to fruition, and the questions below may serve as an aid towards this goal. 1 - The work argues that recourse summaries on sub-groups enable checking models for meaningful and non-discriminatory behavior before deployment. While this is true, it is unclear both qualitatively and quantitively whether and to what extend summaries are preferred over aggregates of individual recourses. After reading the user-study description in both the main body and appendix, I am left wondering whether the setup is fair. Specifically, it seems that the manner in which the bias was constructed in the data very closely mirrors that type of two-level explanations that the proposed approach, AReS, will be outputting. Thus, I am not sure whether the presented comparison is fair. 2 - It is unclear why the analysis is restricted to two-levels? Why not more? Also, why not just one? For instance, in the example of Figure 1, I would imagine an interesting multi-level query would involve foreign workers who have work visas vs student visa vs refugees. The current two-level setup wouldn't seem to allow for this directly, unless you define the outer level as "foreign and work visa", etc. If so, this begs the question of why would we need two-levels in the first place, if we can define any sub-group as (an albeit longer) conjunction of predicates defining the members of that subgroup. 3 - On recourse cost, in line 179 it is unclear how the proposed approach would handle categorical variables; relevantly the datasets used do not seem to include categorical variables (e.g., the Adult dataset has a number of categorical features). Furthermore, it is peculiar why the authors would not use the {max,log}-percentile shift cost functions defined in [31] as it captures their motivation on line 177 fairly well. Could this choice be due to an undisclosed restriction of the optimization procedure? Moreover, in line 193, the point about uncertainty in actionability seems to be abandoned in the remainder of the paper. Finally, and perhaps critically, the discussion of cost would benefit greatly from the work of Karimi et al., https://arxiv.org/abs/2002.06278, where it is argued that cost of guaranteed recourse should only be defined when considering the causal relations between variables. Although, I should point out that defining cost functions is an open problem, and commend the authors work for bravely taking on such a challenge. I am simply pushing for a more transparent discussion on the limitations of cost functions, and believe that the community would benefit from not sweeping unresolved challenges under the rug.

Correctness: 4 - In Table 2, LR methods show 100% recourse accuracy for the AR algorithm, by design. Building on the theory developed in this work, they show that, subject to actionability constraints, their returned flip sets are optimal. Thus, the Mean FCost column showing lower cost for AReS as compared to AR is misleading and incorrect. This is likely an artifact of the parameters used for running AR, which if I recall correctly, the default bin discretization is set to 100. I suppose that increasing the bins would lead to lower cost, which again, based on the theory in their paper [31], should result in optimal recourse actions. Also in Table 2, it is peculiar why the authors compare AReS with [22,31] but not with [10], which is by design model-agnostic and by theory optimal, and would thus not suffer from the limitations mentioned in line 323. This leaves me wondering whether and to what extend the statement in line 317 about lack of baseline holds, and whether the contribution of this work competitively extends to the individualized recourse setting. This is why I believe that the primary contribution is in actionable recourse *summaries* for sub-groups, but not necessarily individual recourse.

Clarity: Yes.

Relation to Prior Work: Yes. I should point out, that while I appreciate the authors' efforts to relate their proposed formulation to previous work (in the Appendix under "Unifying Prior Work"), unfortunately, I am failing to see how this is different that writing the Lagrangian form of a general constrained optimization formulation for counterfactual explanations. (see for e.g., Dandl et al., https://arxiv.org/pdf/2004.11165.pdf) Perhaps I am not seeing the novelty here, and would very much appreciate correcting my understanding.

Reproducibility: Yes

Additional Feedback: Scatterd nits; - [line 30] as pointed out by the authors, the approach of [33] is gradient-based, and thus by definition not model-agnostic (e.g., cannot work for a decision tree) - [line 95] missing relevant citations Barocas et al., https://dl.acm.org/doi/abs/10.1145/3351095.3372830, Venkatasubramanian et al., https://dl.acm.org/doi/abs/10.1145/3351095.3372876, Karimi et al., https://arxiv.org/abs/2002.06278 - [line 215] in the optimization problem, the constraint set "C" is undefined - [line 335] the authors point out that humans are able to comprehend and reason about explanation sizes of around 10-15. This argument seems to contrast the very well-known work of George Miller [1956] on "The Magical Number Seven, Plus or Minus Two: Some Limits on Our Capacity for Processing Information". ------------------------------------------------------------------------------------ Post-rebuttal update: Having read the other reviews and the authors' responses, my score remains as before. Allow me to explain my reasoning for the respective authors: On the one hand, I agree with the novelty of the paper's contributions, as: 1. the contents constitute a new avenue of research in sub-population based recourse explanations; and 2. the authors present a good first attempt at a technical solution to this formulation; thus, the work leaves much room for future research and improvement, which is welcoming; and 3. the authors bravely tackle the problem of defining cost, and present a new (yet not proven and not empirically demonstrated; see R3 comment) cost approach On the other hand, I would like to echo again that several statements seem to over-claim contributions and/or sweep assumptions/corner-case restrictions under the rug. For instance: - See my comment (under Weaknesses) about why they considered two-level sets and not more (the authors' response is not entirely convincing) - See my comment (under Relation to prior work) about "unifying prior work" - See R2's comment (under Weaknesses) about using recourse for bias detection and the setting that the authors don't consider - See R3's comment (under Weaknesses) about interactions in the suggested recourse (to which the authors responded that is unlikely but possible) - See R3's comment (under Correctness) about empirical results Overall, I think the paper touches on a number of topics (a new recourse definition, an attempt at discovering cost, optimization difficulty, assaying fairness), and given the limited page count, I understand that only so much can be said about each topic. With that said, if the paper were to be accepted, I would suggest that the authors: a. more clearly highlight its shortcomings so the community can unambiguously focus on building in those directions b. perhaps move the fairness section to the appendix, leaving more room in the main body to clarify the questions above c. add (and incorporate) missing citations as pointed out by R2 and myself Best of luck!


Review 2

Summary and Contributions: This paper proposes a methodology for creating a small number of global-level counterfactual explanations that provide useful recourse for as much of the population as possible. Furthermore, it proposes a framework to learn costs from expert or user input, in contrast with previous methods that seek to estimate these costs purely from the data distribution. The method is then evaluated on real world datasets. Finally, the paper presents a user study demonstrating the utility of the proposed methods in detecting bias in the underlying classifier. Update after rebuttal: I appreciate the comments from the authors. If accepted, I'd recommend the authors take seriously the reviewers' comments about overclaiming. In general, I'm satisfied by the author response.

Strengths: This work has a number of elements that I find innovative: - A meaningful definition of global-level counterfactual explanations. - An attempt to explicitly learn costs from user input. - An analysis of the trade-offs between interpretability (size of explanations) and accuracy. I think it's of core relevance to the conference, building on a growing line of work in this area.

Weaknesses: The main drawback to this paper is that because the proposed techniques differ significantly from existing methods, many of the metrics and objectives here are heuristics, and they can be somewhat arbitrary. This is not to say that they can't be useful, but it's not always clear why certain choices are made. For example, in modeling costs, the authors propose both featurecost() and featurechange(), where feature cost represents a fixed cost for each feature and featurechange captures the magnitudes of these changes. It's not clear to me why these are both necessary, especially because most prior work on counterfactual explanations/actionable recourse consider a single cost function that captures the cost of moving from one point to another. Some justification here would help. The discussion of discovering bias also seems a bit tenuous. As the authors point out in the broader impacts section, the proposed techniques could lead to spurious claims of bias that arise from different data distributions. One thing that might strengthen this is a discussion of the direction of the errors: does this method always skew towards indicating bias, even when it isn't present, or does it also miss clear-cut cases? This may be fairly hard to define or make strong claims about, but if possible, I think it would give a better sense for the utility of this method.

Correctness: The claims and proofs appear correct.

Clarity: The paper is really well-organized, making it quite easy to read. Section 2, in particular, did a good job of streamlining and presenting a lot of information.

Relation to Prior Work: The paper clearly differs from related work. In addition to the prior work mentioned in the paper, the following papers provide critiques of counterfactual explanations and actionable recourse. I'd be interested in seeing a discussion of how this work engages with those critiques. - "The hidden assumptions behind counterfactual explanations and principal reasons" (Barocas et al., 2020) - "The philosophical basis of algorithmic recourse" (Venkatasubramanian and Alfano, 2020)

Reproducibility: Yes

Additional Feedback: - line 198: p_{i,j} should be p_{ij} for consistency - line 283: double reference [22]


Review 3

Summary and Contributions: This paper constructs two-level rule lists that provide users with recourse actions for a black-box model. The top level is based on demographic information and the second level are regular rule lists. Experimental analysis is based on accuracy (whether the recourse actions provided would actually change the model's decision if followed) and mean feature cost (the average cost to change the suggested features for the recourse actions that are accurate). These experiments show that these two-level recourse summaries have high accuracy while maintaining low feature cost across a number of datasets and models. A small user study finds that the summaries are interpretable when focused on a fairness-related task.

Strengths: The authors introduce a new technique that builds well on previous research to address a previously unmet need that is well-grounded in an understanding of fairness-related and other societal needs when dealing with black-box models. The results show that this new method is accurate and usable. I could see this tool as genuinely useful to practitioners.

Weaknesses: These are really questions and potential concerns, not necessarily weaknesses. Feature cost calculations: One of the interesting things about the introduced AReS framework is the idea that feature costs can be learned via pairwise comparisons. Yet in the evaluation section these feature costs are just randomly distributed, if I understand correctly. Why not use a user study to evaluate this part of the framework as well? And - assuming I understood correctly that the feature costs were uniformly randomly distributed - does the experimental evaluation change under non-uniform feature cost choices? This seems like a critical part of the evaluation that is currently missing. Recourse interactions: Did you study the possibility of interactions in the suggested recourse (within a second-layer per-demographic set)? I.e., is there a possibility that one of the suggested recourse rules could contradict another one? POST-REBUTTAL UPDATE: The authors refer to some interesting results in their rebuttal addressing the above two points, e.g., that recourse interactions don't happen frequently in practice. It would be good to incorporate these results into the main text. Additionally, and importantly, the authors do not address their issue of over-claiming. For this paper to be accepted, the authors must agree to add a clear Limitations section that describes honestly and openly the problems with their work, including those mentioned above in this review as well as by the other reviewers.

Correctness: The main concern is above. One other issue that can be easily corrected: on page 7 the authors state "Our method, on the other hand, consistently obtains the most accurate as well as the lowest cost recourses across all datasets and with all black box models." This is not precisely true and there is no need to over-claim strong results here. Please restate this sentence to indicate that AReS generally does well on both accuracy and feature cost across all datasets and models (but do not claim that it is always the best).

Clarity: Yes.

Relation to Prior Work: Yes.

Reproducibility: Yes

Additional Feedback: Details: Line 111: "criterion" and the rest of the sentence should be made plural Line 283: [22] is cited twice in a row


Review 4

Summary and Contributions: This paper proposes a framework of Actionable Recourse Summaries, to construct counterfactual explanations. The method can provide interpretable and accurate summaries of cost-effective recourses for the entire population with emphasis on specific subgroups of interest. Experiments with 3 datasets show the potentials of the method to diagnose model biases and discrimination.

Strengths: Along with the proposed Actionable Recourse Summaries method, the paper also established a set of metrics for the optimization: Interpretability, and Recourse Correctness, Coverage and Costs. Expirical results show that with comparable recourse accuracy, the proposed method has the lowest feature cost compared to baselines. A user study and qualitative examples (in the appendix) further demonstrate how baises in recourses are detected.

Weaknesses: The proposed method is shown to work with tabular data, but it's not clear how to adapt it to other formality, such as visual or textual data. Also, it'd be nice to touch the base on complexity/computation analysis of the proposed method against baselines.

Correctness: I think the empirical methodology is correct, but didn't proof check all the theorems.

Clarity: The paper is well written.

Relation to Prior Work: Relation to prior work is clearly discussed.

Reproducibility: Yes

Additional Feedback:

[Author Response · NeurIPS 2020]

We thank the reviewers for their insightful feedback. We will incorporate all the suggestions/clarifications in the final
version. Our detailed comments are provided below. All references are to the citations in the submission.

**Novelty and Contributions**: While there has been a lot of prior work on generating individual recourses via local
counterfactual explanations, there is little to no work on *global* counterfactual explanations which can provide a high-
level summary of recourses associated with a given (black box) model. This work makes the first attempt at addressing
this critical gap. Our main contributions are: 1) We introduce the notion of **global counterfactual explanations** and
propose the first framework, AReS, to generate them. Our explanations provide interpretable, customizable, and accurate
summaries of *actionable* recourses for the *entire population* with emphasis on specific subgroups (which are either
input by end users or learned automatically). 2) Our work also outlines one of the first solutions for **learning feature
costs** from user inputs on pairwise feature comparisons. While we demonstrate (Theorem 2.2) that our optimization
problem reduces to the generalized constrained optimization formulation for local counterfactuals [31,33] and can
thereby generate individual recourses as well, the main goal of this result is to establish connections with prior research
and not to suggest that generating individual recourses is one of the main contributions of our work.

**Feature costs and feature changes**: We consider two kinds of recourse costs: `featurecost` which captures the notion
that some features can be intrinsically harder to change than others; and `featurechange` which captures the notion
that changing feature values gets harder as the magnitude of the change increases. Depending on the specific application
setting, one of these notions might be more important than the other. So, instead of combining these two notions into a
single cost function, we provide end users with the flexibility to choose their relative importance (by setting $\lambda_3$ and $\lambda_4$).
Our optimization framework is also generic enough to incorporate multiple definitions of the aforementioned costs (e.g.,
`featurechange` can be defined using the percentile shifts in feature values as done in [31]).

**R1**: (i) **User studies**: In addition to the biased two-level model discussed in Section 5, we also experimented with
introducing racial biases into a 3-layer neural network (3-NN) and a logistic regression (LR) model via trial and error.
We then carried out similar user studies (as in Section 5) with 36 participants to evaluate how our explanations compared
with aggregates of individual recourses. In case of 3-NN, AReS clearly outperformed AR-LIME (88.9% vs. 44.4%
on bias detection; 55.6% vs. 11.1% on bias description). In case of LR, AReS and AR-LIME performed comparably
(88.9% in both cases on bias detection; 66.7% vs. 44.4% on bias description). This was omitted due to space constraints,
but will be included in the final version. Also, see *R2: Bias detection* below. (ii) **Two-level decision sets** carry semantic
meaning – with outer level rules describing *subgroups* and inner level rules representing recourses for the corresponding
subgroups. As shown by prior work [14], this interpretation makes it very easy for end users to understand explanations.
Furthermore, our preliminary studies have also shown that users can easily distinguish between subgroups and their
corresponding recourses with two-level rule sets, but experience difficulties in doing so with 1 or $> 2$ levels. (iii) We
account for **uncertainty in actionability** of features by using the *probabilistic* Bradley-Terry model (See defn $p_{ij}$ in
line 195) to learn feature costs. We will make this connection clearer in the final writeup. (iv) **Table 2**: We concur with
the reviewer that our main contribution is recourse summaries. The goal of Table 2 is not to claim that we outperform
individual recourse techniques but to assure the reader that we are not sacrificing recourse accuracy or costs in our
attempt to construct interpretable summaries (as also pointed out by R3). The *mean fcost* metric shows lower values
for AReS compared to AR not due to parameter errors but because the log-percentile shift optimized for by [31] is
different from what is captured by this metric (Lines 308-310). We also compared AReS and AR using the cost function
from [31] and found that AR achieves about 8 to 10% lower costs than AReS, as expected. We will include these
clarifications in the final writeup. (v) **Unifying prior work**: As correctly pointed out, we are just writing down the
Lagrangian form of a general constrained optimization formulation so that it can later be used for proving Theorem 2.2.

**R2**: **Bias detection**: We would like to emphasize that AReS, at its core, is an explainability technique and is not
explicitly optimized for detecting model biases or fairness violations. That said, explainability techniques are commonly
used to detect "potential" model biases or discriminatory behavior [24,33]. The bias detection study (Section 5) is
meant to be a proof-of-concept to demonstrate that AReS can also be used to highlight potential biases, similar to other
explainability techniques.

**R3**: (i) **Feature cost calculations:** We had conducted a user study to obtain pairwise feature comparisons for the
Credit dataset, and had leveraged these inputs to learn feature costs and generate recourse summaries using AReS.
We did not find significant drops/differences in recourse accuracies or mean fcosts (in comparison with the setting
of uniform feature costs). We also experimented with non-uniform feature costs and observed similar results. (ii)
**Recourse interactions:** It is theoretically possible for rules to contradict each other, but in practice we observed this
occurs very rarely ($< 0.2\%$). Our objective already optimizes for *coverage* and *interpretability*, thus providing little
incentive to choose multiple rules that apply to same sets of data points (thereby reducing the chance of contradictions).

**R4**: (i) **Text and image data:** It is easy to extend AReS to domains beyond tabular data, as long as the input features
are interpretable. For example, *bag-of-words* features in case of text, and *super-pixels* of images can be used as inputs
to AReS. Explainability techniques commonly use these kinds of interpretable representations as features [24].

[Meta-Review · NeurIPS 2020]

Thank you for your submission. In discussion following the rebuttal, consensus was reached leaning towards accept. Please incorporate points from the reviewers' detailed comments, your rebuttal and note. In particular, please: clearly highlight limitations tone down claims such as "first ever" incorporate additional references mentioned and brief discussion